# Analysis and Interpretation of Automated Blood Count in the Treatment of Chronic Paracoccidioidomycosis

**DOI:** 10.3390/jof10050317

**Published:** 2024-04-27

**Authors:** Eliana da Costa Alvarenga de Brito, Adriana de Oliveira França, Igor Valadares Siqueira, Vinícius Lopes Teodoro Félix, Amanda Alves Rezende, Bárbara Casella Amorim, Suzane Eberhart Ribeiro da Silva, Rinaldo Poncio Mendes, Simone Schneider Weber, Anamaria Mello Miranda Paniago

**Affiliations:** 1Graduate Program in Infectious and Parasitic Diseases, Faculty of Medicine, Federal University of Mato Grosso do Sul, Campo Grande 79070-900, MS, Brazil; eliana_brito@ufms.br (E.d.C.A.d.B.); dricaseal@gmail.com (A.d.O.F.); babi_casella@hotmail.com (B.C.A.); 2Scientific Initiation CNPq, Faculty of Medicine, Federal University of Mato Grosso do Sul, Campo Grande 79070-900, MS, Brazil; igorvaladares@gmail.com (I.V.S.); viniciuslopex1999@gmail.com (V.L.T.F.); aalvesrezende@gmail.com (A.A.R.); xsuzane@hotmail.com (S.E.R.d.S.); 3Department of Tropical Diseases, Botucatu Medical School, São Paulo State University, Botucatu 18618-687, SP, Brazil; tietemendes@terra.com.br; 4Faculty of Pharmaceutical Sciences, Food and Nutrition, Federal University of Mato Grosso do Sul, Campo Grande 79070-900, MS, Brazil; weberblood@gmail.com

**Keywords:** anemia, blood cell count, leukocyte count, paracoccidioidomycosis

## Abstract

Blood count is crucial for assessing bone marrow’s cell production and differentiation during infections, gaging disease severity, and monitoring therapeutic responses. The profile of blood count in chronic forms of paracoccidioidomycosis (PCM) has been insufficiently explored. To better understand the changes in hematological cells in different stages of the PCM chronic form, we evaluated the blood count, including immature blood cells in automated equipment, before and during the treatment follow-up of 62 chronic PCM patients. Predominantly male (96.8%) with an average age of 54.3 (standard deviation SD 6.9) years, participants exhibited pre-treatment conditions such as anemia (45.2%), monocytosis (38.7%), and leukocytosis (17.7%), which became less frequent after clinical cure. Anemia was more prevalent in severe cases. Notably, hemoglobin and reticulocyte hemoglobin content increased, while leukocytes, monocytes, neutrophils, immature granulocytes, and platelets decreased. Chronic PCM induced manageable hematological abnormalities, mainly in the red blood series. Monocytosis, indicating monocytes’ role in PCM’s immune response, was frequent. Post-treatment, especially after achieving clinical cure, significant improvements were observed in various hematological indices, including immature granulocytes and reticulocyte hemoglobin content, underscoring the impact of infection on these parameters.

## 1. Introduction

Paracoccidioidomycosis (PCM) is a systemic mycosis endemic in Latin America whose etiological agents are fungi of the genus *Paracoccidioides* [1]. Brazil is considered the endemic center of this disease, where it ranks as the eighth leading cause of death resulting from chronic infectious disease [2].

The infection primarily affects the lungs, spreading through the lymphatic and hematogenous routes to any other organs [3]. In most individuals who inhale the fungus, the progression of the infection is contained, and the course of disease will depend on the size of the inoculum, the pathogenicity and virulence of the fungus, the immune response, and possibly genetic factors [4]. Active PCM can be classified into acute/subacute and chronic forms. The acute/subacute form is known as the juvenile form because it affects children, adolescents, and young adults of both sexes aged under 30 years of age, in a similar proportion (male/female ratio of 1.7:1.0). The chronic form affects individuals older than 30 years, usually males (male/female ratio of 22.0:1.0), with a prolonged duration of symptomatology [1]; it predominates in clinical practice, accounting for 75% to 90% of cases [5,6,7].

The host–parasite interactions in PCM and the consequent inflammatory responses of the body can lead to changes in hematological cells, such as anemia, leukocytosis, and eosinophilia [8]. These findings are more commonly reported in the acute/subacute form of the disease [9,10]. The production of cytokines has been implicated in these hematological alterations. In the acute/subacute form, patients often exhibit a mixed TH2/TH9 response, characterized by increased levels of interleukin 5 (IL-5) and IL4. These elevated cytokines can lead to severe eosinophilia and an increase in antibody production by B lymphocytes, respectively [10]. Conversely, individuals with chronic PCM typically demonstrate a TH17/TH22 profile, characterized by heightened production of IL-17, CXCL8, and IL-6 [9,10]. These cytokines play key roles in neutrophil maturation in the bone marrow, the recruitment of neutrophils to the infection site, and the development of anemia, respectively [11,12]. It has been demonstrated that the cytokine pattern influences the outcome of *Paracoccidioides* infection [13].

Few studies on blood cells in chronic PCM have been described, showing changes in the number of erythrocytes and the concentration of hemoglobin [14], as well as in the subpopulations of monocytes and lymphocytes, in untreated patients [15,16,17]. The complete blood count is a simple test performed routinely in most clinics and is important in the clinical management of infectious diseases. It is essential for elucidating whether the bone marrow is correctly performing its functions, with adequate cell production and differentiation, as well as gaging the severity of the condition and the response to treatment. The analysis of immature blood cells in automated equipment is replacing manual counting, becoming possible to assess the involvement of precursor cells in inflammatory and infectious activity, as well as the lack of iron in the body at an early stage. 

Professionals treating patients with chronic PCM should be familiar with the typical results of the complete blood count, its relationship with disease severity, its importance in treatment monitoring, and whether newly introduced parameters in automated complete blood counts can be useful in assessing treatment response. To better understand the changes in hematological cells in different stages of the PCM chronic form, the present study evaluated the blood count before and during the treatment follow-up.

## 2. Patients and Methods

### 2.1. Ethical Aspects

This project was approved by the Committee of Ethics in Research with Human Beings of the Federal University of Mato Grosso do Sul (CEP-UFMS) (number CAAE 21534919.2.0000.0021). All participants signed informed consent forms.

### 2.2. Design, Place, and Period of Study

This was a prospective, quantitative, epidemiological, observational study evaluating the hematological parameters of patients with chronic PCM before the beginning of treatment and during the healing process. 

This study was conducted at the systemic mycoses outpatient clinic of the Infectious and Parasitic Diseases Unit of the Maria Aparecida Pedrossian University Hospital (UNIDIP-HUMAP) of the Federal University of Mato Grosso do Sul (HU-UFMS) in Campo Grande, Mato Grosso do Sul, Brazil, in patients diagnosed between 2013 and 2021.

### 2.3. Inclusion and Exclusion Criteria

Patients of both sexes with proven or probable chronic PCM were included in the study. Exclusion criteria: patients who did not undergo a blood count at the service before starting treatment and those presenting comorbidities of infectious, inflammatory, or neoplastic etiology.

### 2.4. Case Definition

Patients with suggestive clinical manifestations and one of the following findings were diagnosed with PCM: (a) identification of typical *Paracoccidioides* spp. yeast forms in clinical samples by direct mycological examination, culture, or histopathological examination (proven cases); or (b) detection of specific serum antibodies by the double agar gel immunodiffusion test (DID) (probable cases) [18].

### 2.5. Clinical and Demographic Data

For this study, clinical data were obtained from a prospectively collected database. Data related to sex, age, history of professional occupations, lifestyle habits, and clinical information such as disease severity and antifungal treatments were systematically collected at admission and during follow-up. PCM severity was classified as mild, moderate, or severe, as described below [1].

The mild form constituted a decrease in body mass index (BMI) of less than 5% from the usual value and involvement of one or few organs or tissues without functional changes.

The severe form involved three or more of the following criteria: (a) loss of BMI of at least 10%; (b) severe pulmonary involvement; (c) involvement of other organs, such as the adrenal glands, central nervous system and bones; (d) enlargement of lymph nodes in multiple chains in the superficial or deep tumoral form (>2.0 cm in diameter, without suppuration) or in the suppurative form; (e) high titers of anti-*Paracoccidioides* antibodies.

The moderate form was intermediate between the mild and severe forms. 

Laboratory data were obtained at four different stages in relationship to the treatment:

S0—before treatment;

S1—during treatment, at the first visit showing some clinical improvement;

S2—during treatment, at the visit presenting clinical cure (disappearance of signs and symptoms of active disease);

S3—during treatment, at the visit demonstrating serological cure (negative serology by the DID test, maintained for six months) or at least 12 months of clinical cure in patients with nonreactive serum.

The criteria for clinical and serological cure were those defined by Mendes et al. (2017) [1].

### 2.6. Laboratory Procedures

During the stages of the disease, the patients underwent the laboratory tests mentioned below, on a monthly basis until S2, and subsequently every three months until S3.

Vein puncture was conducted employing a vacuum tube, and the duration between collection and examination ranged from 30 min to 4 h.

Complete blood count was determined by the XN 3000-Series hematology analyzer (Sysmex Corporation, Kobe, Japan). Complete blood count parameters of cellular immaturity were evaluated only in patients included after 2017, when the software was available: immature granulocytes (IGs) (myelocytes, metamyelocytes, and promyelocytes); reticulocyte hemoglobin content (Ret-He); the fraction of immature reticulocytes (IRF); and the index of reticulated platelets (RIP). A manual leukocyte differential count was performed when the equipment issued an alert.

The mean corpuscular volume (MCV), mean corpuscular hemoglobin (MCH), and mean corpuscular hemoglobin concentration (MCHC) were also determined.

The reference values adopted for the hematological parameters in this study were in accordance with the specifications of Hoffbrand and Steensma, 2020, and Pekelharing et al., 2010 [19,20] (Appendix A). 

Anemia was considered when the hemoglobin (Hb) level was <13.0 g/dL for male patients and <12.0 for female patients.

In addition to the complete blood count, the patients underwent the DID test using antigens from the B339 strain, as described by [21], for diagnosis and to evaluate the response to treatment, serving as a criterion for serological cure.

### 2.7. Statistical Analysis

Statistical analysis was performed using the Jamovi software (version 1.6) for Windows [22]. To determine whether the continuous variables had a normal distribution, the Shapiro–Wilk test was used. Data that followed a normal distribution are presented as the mean ± standard deviation (SD). Data that did not follow a normal distribution are presented as the median and the first and third quartiles (Q1; Q3). ANOVA and/or Friedman’s test were used to compare the continuous variables between the different stages (S0, S1, S2, and S3). Wilcoxon’s W test was applied to compare continuous variables in two stages (S0 versus S2). For the categorical variables, the Cochran Q test was used to compare the different stages, and the McNemar test was used for the association between two stages, S0 and S2. For the analysis of numerical variables in a single sample, the *t* test for one sample was used. Significance was set up at *p* < 0.05.

## 3. Results

A total of 88 patients were diagnosed with chronic PCM between May 2013 and February 2021, with an average of 7.8 new cases of chronic PCM per year. Of this total, 26 patients were excluded according to the criteria presented, and 62 patients participated in this study (Figure 1). Among these participants, sixty were proven cases, diagnosed through the identification of typical *Paracoccidioides* spp. yeast forms in clinical samples, while only two were classified as probable cases, diagnosed through serology.

The 62 patients studied presented median age of 54 (49.0; 58.0) years old, and 60 (96.8%) of them were men, 55 (90.2%) farmers or former farmers, and 58 (93.5%) were smokers or former smokers; the moderate severity predominated (n = 37; 59.7%) (Table 1). Figure 2 represents the geographic location of the patients’ municipality of residence.

**Figure 1 jof-10-00317-f001:**
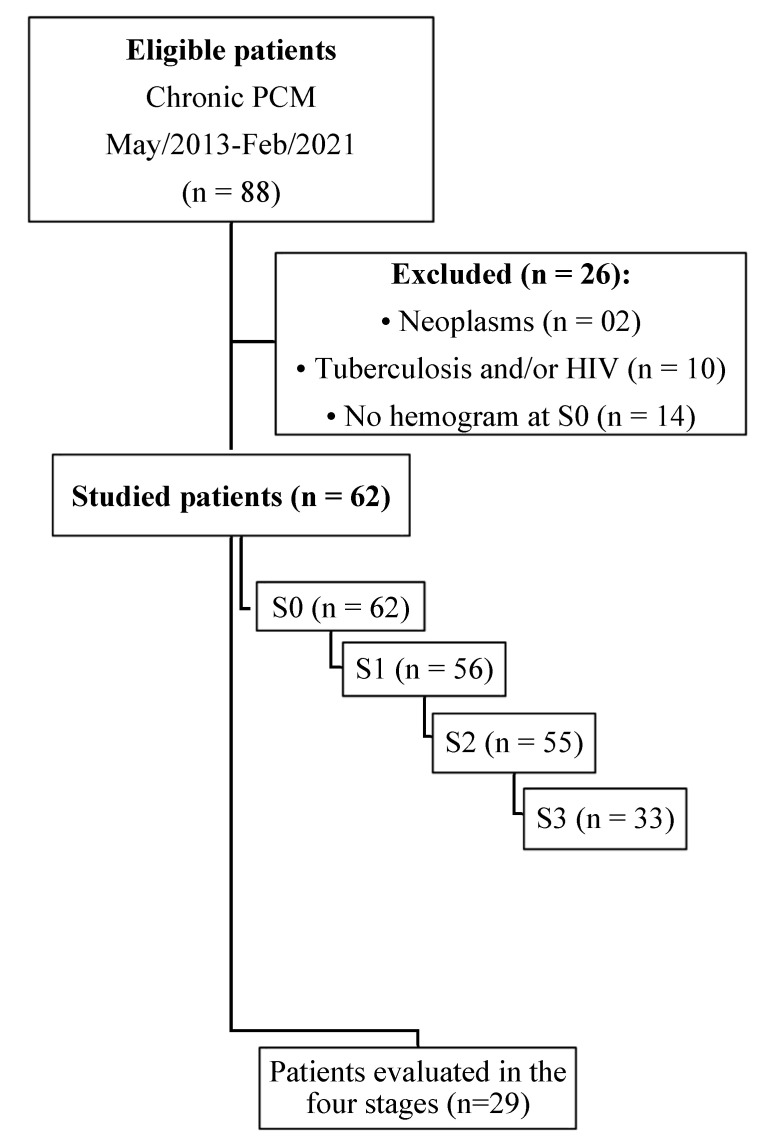
Flowchart of the selection of patients with the chronic form of paracoccidioidomycosis diagnosed at the Federal University of Mato Grosso do Sul. School of Medicine. January 2013–December 2021. n: number of patients; S0: before treatment; S1: between the beginning of treatment and after clinical improvement; S2: clinical cure; S3: serological cure.

**Table 1 jof-10-00317-t001:** Sociodemographic and clinical findings of 62 patients with the chronic form of paracoccidioidomycosis treated at the Maria Aparecida Pedrossian University Hospital between 2013 and 2021.

Variables	n (%)	95% CI	
Sex			
Male	60 (96.8)	88.8	99.6
Female	02 (3.2)	0.4	11.2
Rural activity			
Yes ^‡^	55 (88.7)	78.1	95.3
Never	06 (9.7)	3.6	19.8
Ignored	01 (1.6)	0.0	8.6
Use of Tobacco			
Regularly	51 (82.3)	70.5	90.8
Former smoker	07 (11.3)	4.6	21.9
Never	04 (6.5)	1.8	15.7
Degree of severity of PCM			
Moderate	37 (59.7)	46.5	72.0
Severe	17 (27.4)	16.8	40.2
Mild	08 (12.9)	5.7	23.8
Antifungal treatment			
Itraconazole	39 (62.9)	49.7	74.8
Cotrimoxazole	23 (37.1)	25.2	50.3

CI: confidence interval; n: number of patients; Rural activity: Yes ‡ = individuals who performed rural work in the present or in the past.

**Figure 2 jof-10-00317-f002:**
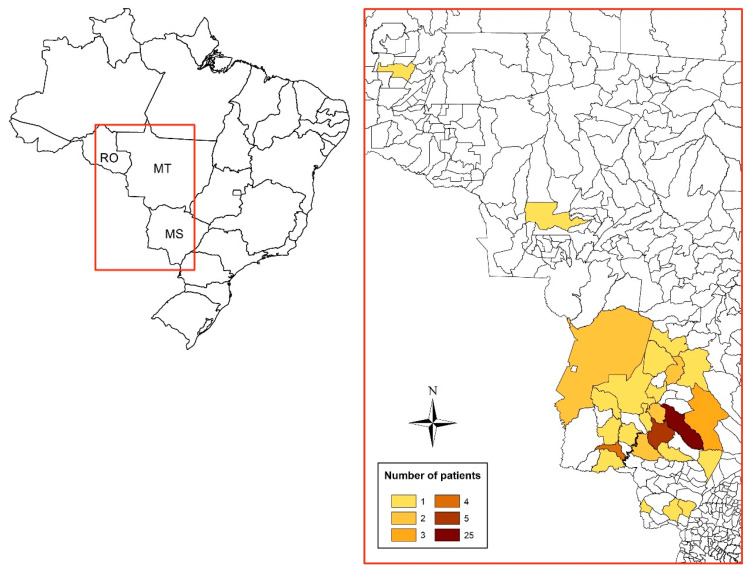
Distribution of residences (municipalities) among 62 patients with the chronic form of paracoccidioidomycosis treated at the Maria Aparecida Pedrossian University Hospital of the Federal University of Mato Grosso do Sul in Campo Grande, Mato Grosso do Sul, Brazil. RO: Rondônia; MT: Mato Grosso; MS: Mato Grosso do Sul.

During the treatment follow-up, not all patients showed compliance with the appointments. The median (first quartile; third quartile) time for patients to reach stage 1 was 2 months (1; 3), that for stage 2 was 5 months (4.0; 8.5), and that for stage 3 was 16 months (13; 25).

### 3.1. Red Blood Series

The analysis of the red blood series before treatment showed that 58.1% of the patients had at least one change. At this stage, anemia was observed in 28 (45.2%) patients, 21 (75%) of whom presented the normocytic and normochromic type. 

The follow-up of 55 patients showed that the frequency of anemia decreased at S2 (*p* = 0.008), while the evaluation of 29 patients revealed a tendency to reduce this prevalence along stages S0–S3 (*p* = 0.063) (Table 2). The progress of erythrocytes, hemoglobin, hematocrit, MCV, MHC, and CMHC was very similar to that of anemia, except the erythrocyte count that showed no difference along stages S0–S3 (Figure 3 and Appendix A).

**Table 2 jof-10-00317-t002:** Anemia observed before treatment and its progress after introduction of the antifungal compounds.

Variable	Patients (n)	S0 n (%)	S1 n (%)	S2 n (%)	S3 n (%)	*p* Value
Anemia	62	28 (45.2)	...	...	...	...
	55 *	21 (39.3)	...	11 (20.0)	...	0.008
	29 **	08 (27.6)	06 (20.7)	06 (20.7)	03 (10.3)	0.063

n: number of patients; …: not performed; S0: before treatment; S1: between the beginning of treatment and after clinical improvement; S2: clinical cure; S3: serological cure. Statistical analysis: McNemar test *; Cochran’s Q test **.

**Figure 3 jof-10-00317-f003:**
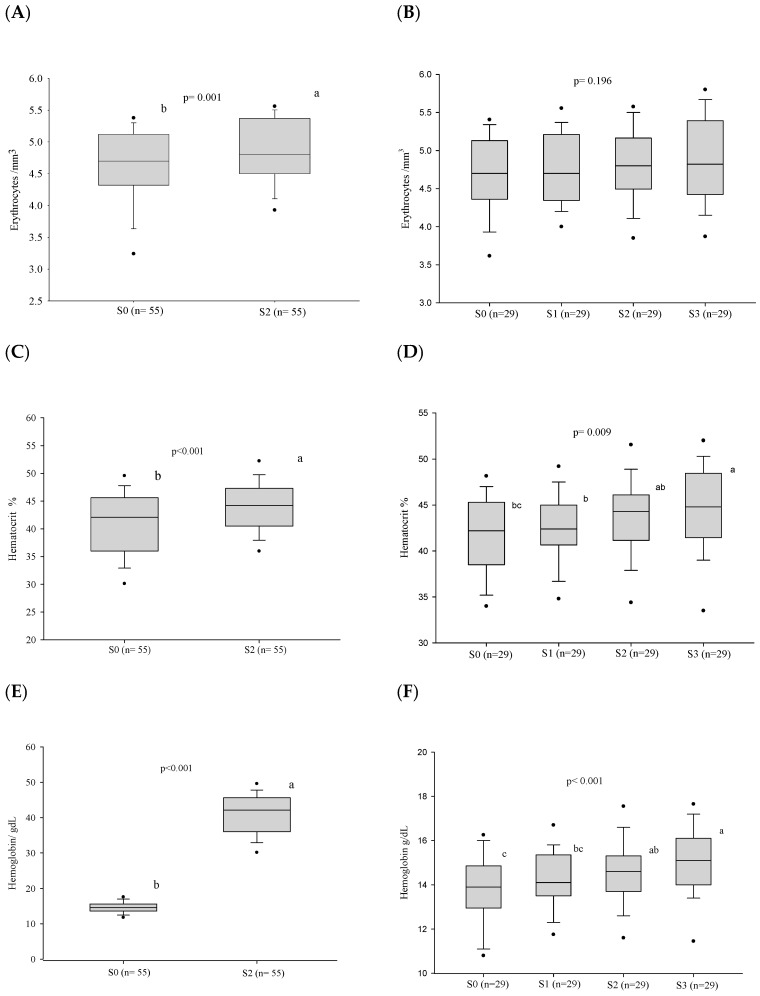
Blood red cell parameters before treatment and progress after introduction of the antifungal compounds. (**A**) Comparison of the erythrocyte values before treatment and at clinical cure. (**B**) Comparison of the erythrocyte values between the different stages. (**C**) Comparison of the hematocrit values before treatment and at clinical cure. (**D**) Comparison of the hematocrit values between the different stages. (**E**) Comparison of the hemoglobin values before treatment and at clinical cure. (**F**) Comparison of the hemoglobin values between the different stages. n: number of patients; S0: before treatment; S1: between the beginning of treatment and after clinical improvement; S2: at clinical cure; S3: at serological cure. Statistical analysis: Wilcoxon W test; Friedman test—post hoc (Durbin–Conover). Lower-case letters compare medians; different letters are statistically significant differences (*p* ≤ 0.05), while medians followed by the same letter or not followed by any letter do not differ (*p* > 0.05).

### 3.2. White Blood Series

#### 3.2.1. Frequency of Alterations

At admission, before the introduction of the treatment, 50 (80.6%) patients presented at least one alteration of the white blood series—leukocytosis: 11 (17.7%); monocytosis: 24 (38.7%); lymphopenia: 19 (30.6%); eosinophilia: 15 (24.2%); neutrophilia: 8 (12.9%).

After treatment, the frequency of neutrophilia and monocytosis showed a decrease at the stage of clinical cure (S2) in the 55 cases evaluated (Table 3). However, the frequency of hematological alterations showed no differences in stages S0–S3, in the 29 patients under follow-up. A tendency to decrease was observed only with monocytosis (*p* = 0.059) (Table 3).

#### 3.2.2. White Blood Cell Counts

After treatment, a decrease in the leukocyte, neutrophil, and monocyte counts was observed at the stage of clinical cure (S2) in the 55 cases evaluated. (Figure 3 and Appendix A). In addition, the neutrophil/lymphocyte ratio and the monocyte/lymphocyte ratio also decreased in the same period. However, no alteration of the cell counts was observed in stages S0–S3 in the 29 patients under follow-up. Nevertheless, the monocyte/lymphocyte ratio decreased in the period S0–S3 (Figure 4 and Appendix A).

### 3.3. Platelet Series

At admission, before treatment, thrombocytosis was present in 9.7% of the patients. During the follow-up, a decrease in the platelet count and platelet/lymphocyte ratio was observed both at clinical cure, when 55 patients were evaluated, and at serological cure, in the evaluation of 29 patients (Figure 5 and Appendix A). 

### 3.4. Cellular Immaturity

The cellular immaturity, evaluated in the granulocytes and in the red blood series, showed a decrease in the percentage of immature granulocytes (IGs), an increase in the reticulocyte hemoglobin content (Ret-He), and an increase in the reticulocyte count (IPF) after reaching clinical cure at S3 (Table 4).

### 3.5. Influence of the Severity of Paracoccidioidomycosis in Patients on Hematologic Alterations

Patients with the severe clinical form showed a higher frequency of anemia than those with the mild and moderate forms, taken together (Table 5).

### 3.6. Influence of the Antifungal on the Hematologic Alterations

The variances noted in erythrocytes, leukocytes, neutrophils, platelets, IG, and Ret- He between stages S2 and S0 were not linked to the specific antifungal employed, be it cotrimoxazole or itraconazole (Appendix A).

## 4. Discussion

Among the complementary laboratory tests, the complete blood count is one of the most requested in clinical practice and is essential in the diagnosis and control of infectious diseases. Abnormalities in blood cellularity before treatment result from the action of the fungus and/or the inflammatory response of the host, and it is expected that with treatment, they will return to normal. Understanding that blood counts aid in the clinical management of PCM, we present the analysis and interpretation of the blood counts of patients with the chronic form of the disease before treatment and during follow-up. This study evaluated the complete blood count and parameters of cellular immaturity: immature granulocytes (myelocytes, metamyelocytes, and promyelocytes) and the reticulocyte hemoglobin content to assess functional iron deficiency and reticulocyte production, and the index of reticulated platelets to evaluate thrombopoiesis.

The demographic, epidemiological, and clinical characteristics of the participants were similar to those observed in large series of PCM in its endemic area [4,5]: males, farmers or former farmers, and smokers, with a predominance of mild/moderate PCM and the mean age in the fifth decade. This similarity suggests that our findings may be applicable to a broader population of PCM patients. Given that age and sex can influence blood cell counts, we relied on reference values for adults categorized by sex [19]. However, it is important to note that nearly all participants were smokers, and research suggests that smokers tend to have elevated levels of various blood cells, including leukocytes, erythrocytes, and platelets [23].

Biological criteria are used to identify when PCM has been cured. Thus, in this study, the different stages of PCM treatment were considered instead of a pre-established time. Improvement in symptoms was observed after approximately two months, reaching clinical cure at about five months after treatment. A similar time interval was also observed in another study [24].

Before treatment, erythrocyte, Hb, and hematocrit (Hct) values were below the reference values in most patients, which may be related to the activation of the immune system by the infectious process, with the release of cytokines such as tumor necrosis factor-α (TNF-α), interferon-γ (IFN-γ), interleukin 6 (IL-6), and interleukin 1 (IL-1). These cytokines inhibit erythropoietin, leading to inadequate production of bone marrow cells, activating phagocytic cells that remove circulating erythrocytes, decreasing their lifespan, and inducing the production of hepcidin and lactoferrin, which promote iron retention in the phagocytic mononuclear system [11]. Infectious and inflammatory processes with a longer course most often progress with the onset of anemia, known as anemia of inflammation (AI) [25]. PCM may also present with normal or low serum iron levels, normal or slightly increased ferritin, transferrin saturation > 15%, and high C-reactive protein (CRP) [26] .

Biomarkers indicating the severity of PCM can greatly improve the clinical management of the condition, aiding in decisions regarding hospitalization and treatment selection. Currently, severity assessment primarily relies on clinical criteria, except for elevated antibody titers detected by DID [1]. Our findings demonstrate an association between anemia and PCM severity, prompting attending physicians to consider its presence as a potential indicator of the case’s severity.

Several systemic infections caused by viruses, bacteria, protozoa, or fungi can lead to the emergence of AI [27]. Most tuberculosis patients showed hematological and inflammatory profiles consistent with AI, predominantly of the normochromic and normocytic type [28,29]. Our study showed similar results; anemia was observed in almost half of our patients at diagnosis, with a predominance of the normocytic normochromic type. After antifungal treatment, Hb and Hct increased and anemia became rarer. Considering that anemia in PCM is, in general, an AI, the analysis of ferritin and transferrin saturation is recommended before starting iron supplementation since it can be harmful to the patient by favoring fungal growth, which depends on iron for its metabolism [30].

In automated equipment, the Ret-He parameter is an index that indicates the onset of iron deficiency in the body, assessing its incorporation into reticulocytes and contributing to the analysis of its status. Under conditions in which Hb synthesis is compromised, such as in AI, Ret-He may decrease [31]. In the present study, the mean values significantly increased with treatment.

Some leukocyte alterations were observed before treatment, especially monocytosis. Leukocytosis and neutrophilia were found only in a small percentage of patients. The defense mechanism against infection is not simple and requires the involvement of many cells of the immune system, such as phagocytic cells—monocytes and neutrophils—which are the first line of defense of the body against infection by the fungus [32,33]. The role of neutrophils in PCM is still not entirely clear because while its phagocytic function is preserved, they fail to digest both *P. brasiliensis* yeast forms and conidia [34,35]. Although these cells are the first in the process of diapedesis to the site of infection, in PCM, a persistence of neutrophils is observed in the tissue lesions caused by the fungus in the later stages of the disease. This fact has demonstrated the presence of neutrophils around the fungus in the chronic phase of PCM in infected mice [36] and in biopsied tissue [37]. Neutrophils also play a key role in the acquired immune response through the synthesis of pro-inflammatory cytokines, which help to mount a more efficient adaptive immune response [38].

IGs, neutrophil precursor cells evaluated by automated blood count, were not above normal values before treatment but showed a decrease upon clinical cure, suggesting involvement in the response to paracoccidioidal infection.

Although monocytosis was observed in less than a half of the patients before treatment, the blood monocyte count decreased after the introduction of the antifungal compound. A previous study of patients with chronic PCM demonstrated that the blood monocyte count was higher than that of healthy individuals [16]. These findings demonstrate that these cells play a role in the response to paracoccidioidal infection, confirming previous findings: monocytes and macrophages play a key role in phagocytosis, in both the innate and acquired immune response, and are activated by cytokines, which can control the multiplication of the fungus [39]. In addition, blood monocytes are attracted into the infected tissue by *Paracoccidioides* spp., where they undergo changes but carry out anti-paracoccidioidal activities and the immune response [40]. 

Alterations in the lymphocyte count were few, with a small percentage of lymphopenia and no lymphocytosis. However, an increase in the lymphocyte count was observed after specific antifungal treatment. These findings suggest that the functional compromise could be more relevant than numerical ones. In addition, the few cases of lymphopenia in chronic PCM could be related to the reduction in the CD4^+^ subpopulation in the blood and the high concentration of these cells in tissue granulomas [32,41].

The occurrence of eosinophilia in PCM patients is related to the high production of IL-5 due to the activation of the Th2 subpopulation [32], and has been reported mainly in the acute/subacute form [42,43]. As this study focused predominantly on cases of the mild or moderate chronic form, this finding was not awaited. In addition, intestinal parasitic diseases, which are known causes of eosinophilia—especially strongyloidiasis—have been frequently associated with PCM [7] but are not routinely investigated.

Thrombocytosis was only observed in less than 10% of the patients before treatment, but the decrease in platelet counts after treatment suggested an interference of PCM with this finding. Gorelik et al. (2017) [44] observed that the increased synthesis of the cytokines IL-3, IL-6, IL-11, and IFN-γ led to increased production of platelets in the bone marrow due to the stimulation of thrombopoietin. Platelets are also found at the site of tissue injury and play a role as immune cells in the parasite–host interaction, leading to platelet activation. This activation induces the release of several molecules, such as chemokines, cytokines [45], and proteins with antimicrobial action [46]. In addition, platelet activation leads to an increase in the adaptive immune response by activating phagocytic cells and their recruitment to the site of infection [45,47]. 

Due to the instability of leukocyte cells in infection/inflammation, the ratios between cells such as neutrophils/lymphocytes (NLR), monocytes/lymphocytes (MLR) and platelets/lymphocytes have been used as prognostic biomarkers in neoplasias [48] and in systemic inflammation [49]. Recently, in PCM caused by *P. lutzii*, the association of the NLR with disease severity was reported, showing that it can be used as a biomarker of severity in PCM [50]. In the present study, the NLR, MLR, and PLR were evaluated, but no association was found between the cellular ratios and the severity of chronic PCM. However, there was a significant reduction in the healing process in MLR and PLR, reflecting the reduction in monocytes and platelets after treatment starting, respectively.

In severe chronic forms, as well as in the acute/subacute form, the TH2 response is exacerbated, leading to increased IL-4 levels that stimulate antibody production by B lymphocytes [10]. Consequently, antibody titers detected by immunodiffusion have been utilized as severity biomarkers [1]. However, immunodiffusion is not widely performed in laboratories, lacks standardization, and requires approximately ten days to obtain results [51]. Identifying blood cell count alterations as potential severity biomarkers in PCM would be invaluable. However, apart from anemia, no other alterations have been shown to be associated with disease severity. It is noteworthy that cytokines present in the chronic form of PCM contribute to both increased production in the bone marrow and the mobilization of cells to the site of infection. Therefore, these cytokines may not significantly alter absolute or relative cell numbers in circulating blood.

Table 6 summarizes our main findings regarding complete blood cell counts in active chronic PCM and their probable mechanisms. 

The anti-folate effect of cotrimoxazole, primarily due to trimethoprim, can induce megaloblastic alterations. Prolonged or high-dose administration of cotrimoxazole has been associated with macrocytic anemia, mild thrombocytopenia, and leukopenia, attributed to diminished folate levels available for hematopoiesis [54]. In our study, the changes in erythrocytes, leukocytes, neutrophils, and platelets observed after initiating treatment, upon clinical cure, were not significantly greater in those who received cotrimoxazole compared to those who received itraconazole. 

The main limitation of this study was the relatively low number of patients studied because of the low annual incidence of PCM [2]. Nevertheless, this is one of the few publications on hemograms that evaluated the data during the patient’s follow-up after treatment. 

Another limitation of this study was the inability to obtain reference values from a healthy population in the same geographical region as the study patients. Consequently, the frequencies of hematological alterations observed may have been either underestimated or overestimated, as these values are influenced by geographical origin, reflecting the genetic aspects and dietary habits of the population [55]. Although information regarding patients’ dietary habits was not gathered in this study, a previous investigation within the same medical service found comparable daily intakes of energy (Kcal), zinc, iron, magnesium, and copper among PCM patients and healthy controls [14]. However, it is important to note that these factors do not impact the absolute values and dynamics of the parameters under study.

## 5. Conclusions

Alterations in blood cell count seem to be neither frequent nor intense upon admission of PCM patients with the chronic form, characterized by anemia and monocytosis. When monitoring patients, clinicians should carefully evaluate the blood count and expect that, with effective treatment, there will be an increase in hemoglobin and reticulocytes, as well as a reduction in leukocytes, monocytes, neutrophils, immature granulocytes, and platelets.

## Figures and Tables

**Figure 4 jof-10-00317-f004:**
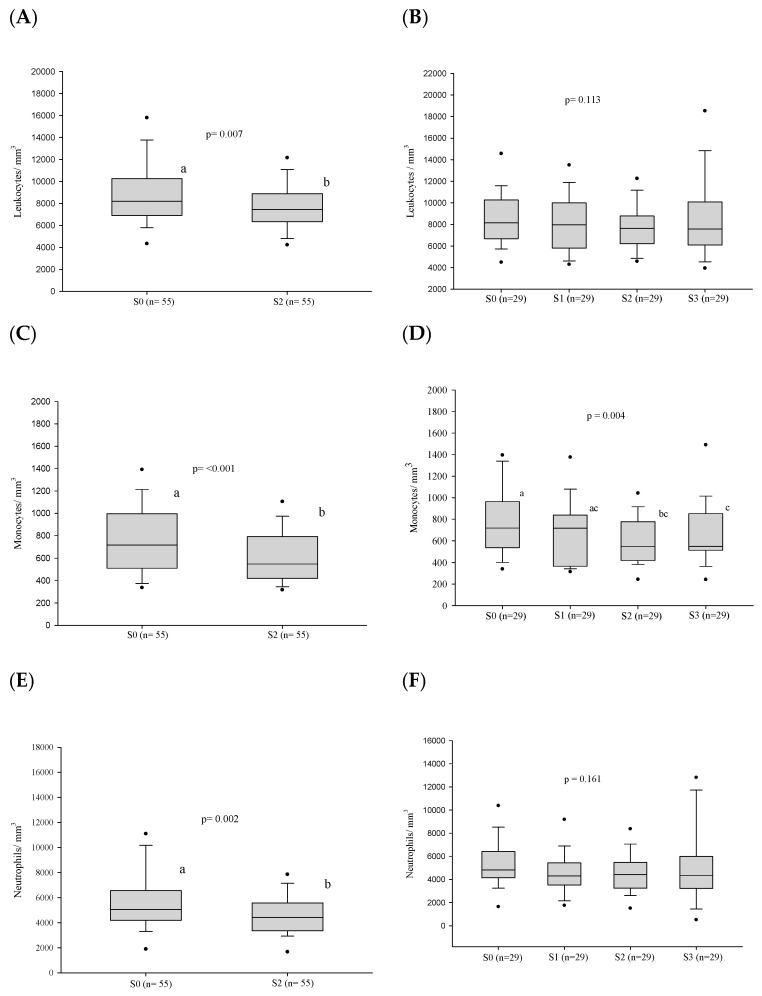
White blood cell count parameters (in number per cubic millimeter) before treatment and progress after introduction of the antifungal compounds. (**A**) Comparison of the leukocyte values before treatment and at clinical cure. (**B**) Comparison of the leukocyte values between the different stages. (**C**) Comparison of the monocyte values before treatment and at clinical cure. (**D**) Comparison of the monocyte values between the different stages. (**E**) Comparison of the neutrophil values before treatment and at clinical cure. (**F**) Comparison of the neutrophil values between the different stages. (**G**) Comparison of the MLR values before treatment and at clinical cure. (**H**) Comparison of the MLR values between the different stages. (**I**) Comparison of the NLR values before treatment and at clinical cure. (**J**) Comparison of the NLR values between the different stages. n: number of patients; NLR: neutrophil/lymphocyte ratio; MLR: monocyte/lymphocyte ratio; S0: before treatment; S1: between the beginning of treatment and after clinical improvement; S2: clinical cure; S3: serological cure. Statistical analysis: Wilcoxon W test or paired Student’s *t* test; ANOVA—post hoc (Tukey), Friedman test—post hoc (Durbin-Conover). Lower-case letters compare medians; different letters indicate that differences are statistically significant (*p* ≤ 0.05), while means followed by the same letter or not followed by any letter do not differ (*p* > 0.05).

**Figure 5 jof-10-00317-f005:**
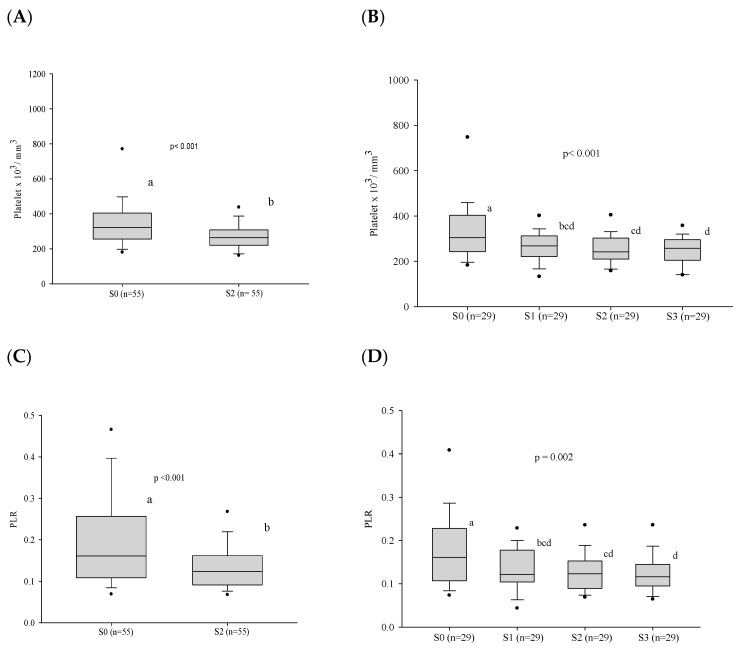
Platelet count and the platelet/lymphocyte ratio before treatment and progress after introduction of the antifungal compounds. (**A**) Comparison of the platelet values before treatment and at clinical cure. (**B**) Comparison of the platelet values between the different stages. (**C**) Comparison of the PLR values before treatment and at clinical cure. (**D**) Comparison of the PLR values between the different stages. n: number of patients; PLR: platelet/lymphocyte ratio; platelet count: number x103/cubic millimeter; S0: before treatment; S1: between the beginning of treatment and after clinical improvement; S2: clinical cure; S3: serological cure. Statistical analysis: Wilcoxon W test or paired Student’s *t* test; ANOVA test—post hoc (Tukey), Friedman test—post hoc (Durbin–Conover). Lower-case letters compare medians; different letters indicate differences that are statistically significant (*p* ≤ 0.05), while means followed by the same letter or not followed by any letter do not differ (*p* > 0.05).

**Table 3 jof-10-00317-t003:** Prevalence of abnormalities observed in the white blood cells before treatment and its progress after introduction of the antifungal treatment.

Variable	Patients (Number)	S0 n (%)	S1 n (%)	S2 n (%)	S3 n (%)	*p* Value
Leukopenia *	55	02 (3.6)	...	02 (3.6)	...	0.564
Leukopenia †	29	01 (3.4)	-	-	02 (6.9)	0.317
Leukocytosis *	55	11 (20.0)	...	08 (14.5)	...	0.366
Leukocytosis †	29	06 (20.7)	05 (17.2)	06 (20.7)	05 (17.2)	0.940
Neutropenia *	55	01 (1.8)	...	03 (5.5)	...	0.157
Neutropenia †	29	01 (3.4)	01 (3.4)	02 (6.9)	03 (10.3)	0.194
Neutrophilia *	55	08 (14.5)	...	04 (7.3)	...	0.206
Neutrophilia †	29	03 (10.3)	02 (6.9)	02 (6.9)	05 (17.2)	0.429
Eosinophilia *	55	14 (25.4)	...	14 (25.4)	...	1.000
Eosinophilia †	29	09 (31.0)	12 (41.4)	10 (34.5)	08 (27.6)	0.274
Monocytosis *	55	22 (40.0)	...	09 (16.4)	...	0.003
Monocytosis †	29	11 (37.9)	06 (20.7)	04 (13.8)	07 (24.1)	0.059
Lymphopenia *	55	14 (25.5)	...	11 (20.0)	...	0.366
Lymphopenia †	29	06 (20.7)	06 (20.7)	06 (20.7)	07 (24.1)	0.954
Lymphocytosis *	55	03 (5.5)	...	03 (5.5)	...	1.000
Lymphocytosis †	29	03 (10.3)	03 (10.3)	02 (6.9)	03 (10.3)	0.801

n: number of patients; …: not performed; S0: before treatment; S1: between the beginning of treatment and after clinical improvement; S2: clinical cure; S3: serological cure. Statistical analysis: McNemar test *; Cochran’s Q test †.

**Table 4 jof-10-00317-t004:** Comparison of the parameters of cellular immaturity in the complete blood count before treatment (S0) and at clinical cure (S2) in 12 patients with the chronic form of paracoccidioidomycosis.

Variables	S0 Mean ± SD Median (Q1; Q3)	S2 Mean ± SD Median (Q1; Q3)	*p* Value
IGs †	0.0 (0.0; 0.1)	0.0 (0.0; 0.1)	0.756
IGs (%) †	0.5 (0.3; 0.7)	0.2 (0.0; 0.3)	0.037
Ret (%) *	1.3 ± 0.6	3.3 ± 8.1	0.942
IRF *	6.8 ± 2.8	7.0 ± 4.0	0.441
Ret-He *	31.4 ± 3.0	33.5 ± 1.7	0.016
IPF †	31.6 (29.5; 33.9)	33.7 (32.8; 34.3)	0.056

SD: standard deviation; Q1: first quartile; Q3: third quartile; IGs: immature granulocytes; Ret: reticulocytes; IRF: fraction of immature reticulocytes; Ret-He: reticulocyte hemoglobin content; IPF: reticulocyte count; statistical test for paired-samples: Student’s *t* test * or Wilcoxon’s rank test †.

**Table 5 jof-10-00317-t005:** Comparison of the prevalence of altered hematological variables and C-reactive protein in 62 patients with the chronic form of paracoccidioidomycosis based on degree of severity.

Variables	Severe (n = 17) n (%)	Mild + Moderate (n = 45) n (%)	*p* Value
Anemia	12 (70.6)	16 (35.6)	0.013
Leukopenia *	0 (0.0)	03 (6.7)	0.555
Leukocytosis	02 (11.8)	09 (20.0)	0.712
Neutropenia *	0 (0.0)	02 (4.4)	1.000
Neutrophilia *	02 (11.8)	06 (13.3)	1.000
Monocytosis	07 (41.2)	17 (37.8)	0.806
Eosinophilia	02 (11.8)	04 (8.9)	0.662
Lymphopenia	02 (11.8)	04 (8.9)	0.662
Lymphocytosis	02 (11.8)	09 (20.0)	0.712
Thrombocytosis *	03 (17.6)	03 (6.7)	0.333

n = number of patients; Statistical analysis: chi-square test or Fisher’s exact test *.

**Table 6 jof-10-00317-t006:** Blood cell count alterations in chronic form of paracoccidioidomycosis: frequencies, mechanisms, and cytokine involvement.

Alteration before Treatment (Frequency)	Mechanism	Involved Cytokines	Related References
Anemia (45.2%)	- Erythropoietin inhibition - Erythrocyte ingestion by macrophages - Iron retention in macrophages	IL-6, IFN-γ, TNF-α, IL-1β, IL-10	[10,11,26]
Monocytosis (38.7%)	- Monocyte activation	TNF- α	[16,52]
Lymphopenia (30.6%)	- CD4+ cell recruitment to the site of infection	IL-17 and IFN-γ	[10,32,41]
Eosinophilia (24.2%)	- Stimulation of eosinophil production	IL-5	[10,13,32]
Neutrophilia (12.9%)	- Stimulation of production in the bone marrow and recruitment to the site of infection	TNFα, IL6, IL1 β, CXCL8	[10,13,53]

## Data Availability

Data are contained within the article and Appendix A.

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
