# Peer review of "Analysis and Interpretation of Automated Blood Count in the Treatment of Chronic Paracoccidioidomycosis"

_jof, 2024, doi:10.3390/jof10050317_

Round 1
Reviewer 1 Report
I consider it an interesting work that provides knowledge to better understand the changes in hematological cells in PCM chronic form
Paracoccidioides must be write in italics in all cases.
Line 41. “The infection primarily affects the lungs, spreading through the lymphatic and hematogenous routes to any other organs” … must be referenced, I advise cite https://doi.org/10.3389/fcimb.2020.605679
Line 83-86. Reference should be made to the criteria used to consider a proven PCM. Use EORT, but since the EORT criteria do not include immunodiffusion, include the global guideline for the diagnosis and treatment of endemic mycoses.
Detection of specific serum antibodies by Immunodiffusion, Which antigen was used?
Laboratory procedures
It is not clear whether a single series of laboratory determinations or more than one was performed, nor at what time after treatment.
Line 166. The acronyms MCV, MHC, and CMHC must be defined.
Table 2, Table 3, Figure 2, Figure 3, Figure 4. Rewrite titles. Results are being presented. If it were an “evaluation”, it should be under discussion.
Conclusion: should be better explained with the analysis and interpretation of the results. Explain more clearly how the evaluation of hematological dynamics can help in the management of PCM patients?
Author Response
Dear reviewer 1,
I would like to express my gratitude for your comments and suggestions for improving the manuscript. The responses are attached herewith.
Best regards.

Reviewer 2 Report
In the manuscript “Analysis and interpretation of automated blood count in the treatment of chronic paracoccidioidomycosis,” the authors analyze and interpret the complete blood count, including immature blood cells, on an automated device before and during follow-up treatment of 62 patients with paracoccidioidomycosis. (PCM), since the hemogram profile in the chronic form of this mycosis has not been sufficiently explored. Their results show that chronic PCM induces hematological abnormalities, mainly in the red blood cell series, and they also showed that monocytosis was a frequent finding, indicating the role of monocytes in the PCM immune response. Furthermore, they showed that after treatment, especially after achieving a clinical cure, significant improvements are observed in several hematological indices, including immature granulocytes and hemoglobin content of reticulocytes, underscoring the impact of infection on these parameters. It is an interesting and well-structured manuscript; however, I have some comments.
In other publications, several authors have emphasized the importance of clearly establishing the reference values of a blood count for well-defined populations, for example, healthy individuals from a specific geographic region, since it has been shown that these values can show differences that can be associated with several factors such as age, gender, geographical origin, among others, so it would be essential to consider the reference values of healthy individuals, in this particular region, to compare them with their results.
On the other hand, the authors should expand the discussion about the sociodemographic factors they obtained. Could these factors affect the blood counts of the patients included in the study?
The authors mention, “To better understand the changes in hematological cells in different stages of the PCM chronic form, the present study evaluated the blood count before and during the treatment follow-up.” With what objective?
Likewise, the authors should discuss how biomarkers of severity in PCM can improve the clinical management of PCM.
On the other hand, the relationship between the cellular and humoral response in patients with PCM determines the disease's prognosis. What advantage would there be in finding biomarkers of severity in blood cell alterations?
Minor fixes:
Line 166: Put the whole meaning of the abbreviations MCV, MHC, and CMHC, which are mentioned for the first time in the text.
Line 511: There should be a corrected space for reference 4.
References
Carefully review all references and strictly follow the journal format.
Author Response
Dear Reviewer 2,
I would like to express my gratitude for your comments and suggestions for improving the manuscript. The responses are attached herewith.
Best regards.

Reviewer 3 Report
Dear Authors,
I read your work on analysing and interpreting automated blood counts to treat chronic paracoccidioidomycosis. The paper, reported as an original article, has caught my attention. The paper is straightforward, easy to read and essential. Moreover, the point is interesting in clinical practice despite the rarity of the disease. The paper could be improved in the introduction and discussion section: considering the exciting results, a more profound discussion would be helpful. Despite that, the paper is well-organized and written.
1) Adding a geographic map of the cases you reported would be beneficial.
2) Lines 50-52 should be improved. Which cytokines or interleukin have been studied in PCM infection related to blood count alterations? Especially in chronic form.
3) It would be interesting to report if some parameters can be used as predictor factors for the outcome or treatment response.
4) Paracoccidioides, italicized in all the text.
5) Patients’ comorbidities and drug intake are missing, excluding neoplastic, infectious and inflammatory diseases.
6) et al., italicized in all the text.
7) What did you use for vein puncture? How many times is intercourse between the puncture and the analytic analyses? Report.
8) “During the treatment follow-up, not all the patients showed compliance with the appointments.” Thank you for reporting this point.
9) In the discussion section, you should report a more critical point of view regarding your results. Although you have reported all the references regarding PCM, it should be exciting and more appropriate to explain why your data are in favour or not with the previous literature or what are the putative molecular or pharmacological reasons of your findings. A table resuming all the main findings will help the reader focus on the clinical step and the analyses to perform in clinical practice, according to your findings.
10) What is your point of view about the point of care in PCM haematological follow-up?
Author Response
Dear Reviewer 3,
I would like to express my gratitude for your comments and suggestions for improving the manuscript. The responses are attached herewith.
Best regards.
